# Secretome of Mesenchymal Stem Cells from Consecutive Hypoxic Cultures Promotes Resolution of Lung Inflammation by Reprogramming Anti-Inflammatory Macrophages

**DOI:** 10.3390/ijms23084333

**Published:** 2022-04-14

**Authors:** Zhihong Xu, Lulu Lin, Yuxuan Fan, Céline Huselstein, Natalia De Isla, Xiaohua He, Yun Chen, Yinping Li

**Affiliations:** 1Department of Pathophysiology, Hubei Province Key Laboratory of Allergy and Immunology, Taikang Medical School (School of Basic Medical Sciences), Wuhan University, Wuhan 430071, China; xuzhihong@whu.edu.cn (Z.X.); linlulu@whu.edu.cn (L.L.); fanyuxuan@whu.edu.cn (Y.F.); hexiaohua@whu.edu.cn (X.H.); yunchen@whu.edu.cn (Y.C.); 2UMR 7365 CNRS, Medical School, University of Lorraine, 54505 Nancy, France; celine.huselstein@univ-lorraine.fr (C.H.); natalia.de-isla@univ-lorraine.fr (N.D.I.)

**Keywords:** inflammation, mesenchymal stem cells, conditioned medium, consecutive hypoxia, macrophages, efferocytosis

## Abstract

The secretome from hypoxia-preconditioned mesenchymal stem cells (MSCs) has been shown to promote resolution of inflammation and alleviate acute lung injury (ALI) through its immunomodulatory function. However, the effects of consecutive hypoxic culture on immunomodulatory function of the MSCs secretome are largely unclarified. Here, we intend to investigate the effects of consecutive hypoxia on therapeutic efficacy of conditioned medium derived from MSCs (MSCs-CM) in alleviating ALI. Human umbilical cord-derived MSCs (UC-MSCs) were consecutively cultured in 21% O_2_ (Nor-MSCs) or in 1% O_2_ (Hypo-MSCs) from passage 0. Their conditioned medium (Nor-CM and Hypo-CM respectively) was collected and administered into ALI models. Our findings confirmed that Hypo-MSCs exhibited increased proliferation ability and decreased cell senescence compared with Nor-MSCs. Consecutive hypoxia promoted UC-MSCs to secrete immunomodulatory cytokines, such as insulin-like growth factor 1(IGF1), IL10, TNFα-stimulated gene 6(TSG6), TGFβ, and prostaglandin E2 (PGE2). Both Nor-CM and Hypo-CM could effectively limit lung inflammation, promote efferocytosis and modulate anti-inflammatory polarization of lung macrophages in ALI models. Moreover, the effects of Hypo-CM were more potent than Nor-CM. Taken together, our findings indicate that consecutive hypoxic cultures could not only promote both proliferation and quality of UC-MSCs, but also enhance the therapeutic efficacy of their secretome in mitigating lung inflammation by promoting efferocytosis and anti-inflammatory polarization of macrophages.

## 1. Introduction

Acute lung injury (ALI) is one of the most common clinical syndromes in critically ill patients. It can result in acute respiratory distress syndrome (ARDS), which is life threatening with high mortality. The main causes of ALI/ARDS involve both pulmonary and extra-pulmonary pathological factors, such as sepsis, severe trauma, and chemical inhalation [1]. Although great progress in the treatment of ALI/ARDS has been made in the past decades, the mortality remains as high as 30–50%. The principal treatments are limited to anti-inflammatory drugs and supportive care, such as mechanical ventilation and extracorporeal membrane oxygenation (ECMO), which may lead to undesirable side effects [2].

Macrophages are the most frequent immune cells in the lung and serve as the first line of immune defense against pathogens. Studies have shown that macrophages are critical in regulating the development of pulmonary inflammation in ALI/ARDS [3]. Upon encounter with foreign stimulators, such as microbes and lipopolysaccharide (LPS), lung monocytes or quiescent macrophages are induced to differentiate into pro-inflammatory macrophages. The pro-inflammatory macrophages initiate inflammatory responses with the release of great amounts of pro-inflammatory factors, which further recruit or activate other immune cells, including neutrophils, lymphocytes, and dendritic cells [4,5]. Overproduction of pro-inflammatory cytokines intensifies the inflammatory injury and triggers cytokines storm, leading to respiratory failure. Limiting excessive pro-inflammatory macrophages while promoting anti-inflammatory macrophages polarization is beneficial to control ALI/ARDS [6]. Although neutrophils are important effector cells to clear pathogens in the early stage of ALI, the lytic forms of apoptotic neutrophils trigger the release of noxious pro-inflammatory cytokines and granular proteins, which worsen lung inflammation and injury [7]. Anti-inflammatory macrophages are essential in clearing apoptotic cells within local tissues, including accumulated neutrophils, through phagocytosis, known as efferocytosis, and therefore promote the resolution of lung inflammation [8,9].

Owing to the potent ability to modulate immune and inflammatory responses, mesenchymal stem cells (MSCs) are emerging as a prospective tool of cell-based therapy in immune disorders as well as inflammatory diseases [10,11]. Studies showed that MSCs could alleviate ALI and ARDS without severe adverse events [12,13]. The therapeutic effects of MSCs are largely owed to their secretome, which is rich in growth factors, cytokines, extracellular vesicles, and exosomes [14,15]. Furthermore, the secretome of MSCs could elicit similar effects or even better effects than their parent MSCs [16,17]. Conditioned medium from MSCs (MSCs-CM) contains the whole secretome of cells, and has been reported to attenuate lung inflammation, promote the functional recovery of multiple sclerosis, and accelerate skin wound healing [18,19,20].

Sufficient cell quantity and optimum cell quality are two essential factors associated with the efficacy of MSCs in cell-based immunotherapy. Cellular senescence is an important cause that inhibits cell proliferation and induces a decrease in cell quality. Senescence of MSCs can not only decrease their viability and expansion efficiency in vitro, but can also mitigate their immunomodulatory capacity, and thus weaken their therapeutic efficacy [21]. Oxygen is a key element for the survival and functional maintenance of cells in tissues. We and others have demonstrated that short-term hypoxia treatment (also known as hypoxia preconditioning) serves as an effective strategy to promote the immunosuppressive capabilities and thus enhance the therapeutic effects of MSCs in multiple diseases [22,23,24,25]. The effects of hypoxia preconditioning on biological functions of MSCs have been extensively explored, including cell proliferation [26,27,28], differentiation [29], paracrine function [19,23,30,31], migration [32], and pro-angiogenesis [30,33], etc. These MSCs pre-conditioned with hypoxia increased migration of human umbilical vein endothelial cells and promoted tube formation of placental microvascular endothelial cells. However, hypoxic pre-conditioning for less than 48 h has been reported to induce MSCs apoptosis [34,35]. As low oxygen tensions between 1–8% is a physiological condition in stem cells niche in vivo [36], long-term hypoxia treatment (or consecutive hypoxic cultures) appears to be more approximate to their physiologic niche than hypoxia preconditioning or normoxic culture in vitro. Nevertheless, the effects of consecutive hypoxia on the immunomodulatory function of MSCs are largely unknown. In addition, the impact of consecutive hypoxia on the proliferation of MSCs remains controversial [27,37]. In this study, we try to investigate the effects of consecutive hypoxic culture on the proliferation and senescence of MSCs, and the efficacy of MSCs secretome in the treatment of ALI.

## 2. Results

### 2.1. Hypoxic Culture Enhanced Proliferation and Inhibited Senescence of MSCs

Human umbilical cord-derived MSCs (UC-MSCs) were consecutively cultured either in hypoxia (1% O_2_) or normoxia (21% O_2_) from passage 0 without interruption. The quantity of MSCs was recorded at the end of each passage. As shown in Figure 1A, the number of MSCs cultured in consecutive hypoxia (Hypo-MSCs) was significantly elevated compared with that of MSCs cultured in normoxia (Nor-MSCs) at each passage. At the end of passage 6, (115.8 ± 11.07) × 10^6^ cells could be harvested from hypoxic cultures, whereas only (55.58 ± 4.22) × 10^6^ cells could be harvested from normoxic cultures. In addition, we also examined the proliferation rate of UC-MSCs in different modes of hypoxia treatment, including normoxic culture, consecutive hypoxic culture, and hypoxia preconditioning. The results demonstrated that the proliferation rate of MSCs from consecutive hypoxic cultures was highest at both 24 and 48 h, and hypoxia preconditioning induced MSCs apoptosis in 24 h (Appendix A). To further assess the quality of cells, we analyzed the level of cellular senescence through senescence-associated β-galactosidase (SA-β-gal) staining. Obvious SA-β-gal^+^ senescent cells were detected in Nor-MSCs of passage 4 and passage 6. Moreover, significantly more SA-β-gal^+^ Nor-MSCs were present in passage 6 than in passage 4. However, under hypoxic cultures, a much lower ratio of SA-β-gal^+^ cells was detected in either passage 4 or passage 6, and no significant difference was found between them (Figure 1B,C). The secretion of senescence-associated secretory phenotype (SASP) in MSCs, including IL6, IL8, and TNFβ, was also reduced by hypoxic culture (Figure 1D), suggesting that consecutive hypoxic culture could attenuate cell senescence and optimize cell quality.

### 2.2. Consecutive Hypoxia Promoted MSCs to Secret High Levels of Immunomodulatory Factors

Previous studies indicated that short-term hypoxia (hypoxia preconditioning) promoted MSCs to secret trophic cytokines and enhanced immunomodulatory capabilities of MSCs. To examine the role of consecutive hypoxia on the paracrine function of UC-MSCs, we analyzed the levels of several immunomodulatory factors in conditioned medium of Nor-MSCs (Nor-CM) and Hypo-MSCs (Hypo-CM), including TGFβ, TNFα-stimulated gene 6 (TSG6), prostaglandin E2 (PGE2), IL10, and insulin-like growth factor 1 (IGF1). Our results showed that Hypo-MSCs produced significantly higher levels of TGFβ, TSG6, PGE2, IL10, and IGF1 than Nor-MSCs (Figure 2). Levels of these factors in MSCs from cultures of hypoxia preconditioning were also analyzed. As shown in Appendix A, hypoxia preconditioning induced a moderate increase in production of TGFβ, TSG6, and IGF1, without altering levels of PGE2 and IL10 when compared to normoxic cultures. However, levels of these factors in MSCs preconditioned with hypoxia were much lower than those in Hypo-MSCs. These results indicated that consecutive hypoxia was more efficient than hypoxia preconditioning in promoting UC-MSCs to produce immunomodulatory factors.

### 2.3. Hypo-CM Promoted Anti-Inflammatory Polarization and Restored Efferocytosis of Macrophages In Vitro

Anti-inflammatory macrophage phenotypes, induced by IL4, IL13, IL10, and glucocorticoids, are critical for controlling inflammation resolution and tissue repair [38]. Since Hypo-MSCs secreted higher levels of anti-inflammatory or immunomodulatory cytokines (Figure 2) and lower levels of pro-inflammatory cytokines (such as IL6 and IL8) (Figure 1D), the effects of Nor-CM and Hypo-CM on polarization of macrophages were analyzed. Bone marrow-derived macrophages (BMDMs) were obtained and stimulated with LPS in vitro, which mimicked the context of ALI. They were first characterized by high expression of F4/80 and CD11b (>85%) (Figure 3A). Upon stimulation with LPS, BMDMs were induced to express a high level of CD80, a marker of pro-inflammatory macrophages. In the presence of Nor-CM or Hypo-CM, the expression of CD80 was significantly downregulated, whereas the expression of CD206, a marker of anti-inflammatory macrophages, was boosted. Moreover, compared with Nor-CM, Hypo-CM led to significantly higher expression of CD206 and lower expression of CD80 (Figure 3B–D). These results suggested that Hypo-CM was more efficient than Nor-CM in promoting polarization of anti-inflammatory macrophages. In addition, Hypo-CM suppressed LPS-induced apoptosis of BMDMs, whereas Nor-CM did not exhibit a similar inhibitory effect (Figure 3E,F).

The shift of macrophages to anti-inflammatory phenotype is accompanied by enhanced efferocytic function [39,40]. We next investigated the effects of Hypo-CM in modulating efferocytosis of BMDMs. BMDMs with different treatments were cocultured with apoptotic Jurkat cells, which were pre-stained by PKH26. Staurosporine (STS) induced significant apoptosis of Jurkat cells (>90%) (Figure 4A,B). As shown in Figure 4C, IL4, which was used as a positive control agent for efferocytosis assay, induced significant engulfment of apoptotic Jurkat cells by BMDMs. In BMDMs stimulated by LPS, only a low level of efferocytosis was detected. However, treatment with Nor-CM or Hypo-CM led to a significant enhancement of efferocytosis in macrophages. The level of efferocytosis in BMDMs treated by Nor-CM was comparable to that treated by IL4. Moreover, Hypo-CM treatment resulted in a higher level of efferocytosis than Nor-CM (Figure 4C,D).

### 2.4. Hypo-CM Effectively Promoted Resolution of Inflammation in ALI Models

To further evaluate the effects of Hypo-CM on the resolution of lung inflammation in vivo, Hypo-CM or Nor-CM was administered into LPS-induced ALI models. Obvious hemorrhage was observed in lungs of ALI models, whereas hemorrhage in animals treated with Nor-CM or Hypo-CM was significantly alleviated, particularly in animals treated with Hypo-CM. Pathological analysis of lung injury with HE staining showed that LPS administration led to remarkable thickening of the alveolar septum with infiltration of inflammatory cells and congestion of venules, as well as the destruction of the alveolus. Both MSCs-CM mitigated the above inflammatory injuries, and Hypo-CM exhibited a better effect in alleviating ALI (Figure 5A,B).

To confirm the effects of Hypo-CM in the resolution of lung inflammation, we analyzed the total number of infiltrated cells and the neutrophils infiltration level in bronchoalveolar lavage fluid (BALF), respectively. In line with the results of pathological injury scores, the number of total infiltrated cells in BALF obviously decreased when mice were treated by either Nor-CM or Hypo-CM, indicating that MSCs-CM limited the infiltration of inflammatory cells in the alveolus (Figure 5C). The infiltration level of CD11b^+^Gr-1^+^ neutrophils in BALF was significantly reduced in mice treated by MSCs-CM (Figure 5D,E). The myeloperoxidase (MPO) activity in lung tissues was also downregulated by MSCs-CM (Figure 5F). In addition, we detected the levels of inflammatory cytokines in the serum of mice. In agreement with the above results, MSCs-CM downregulated the level of IL1β, IL8, and TNFα in the serum (Figure 5G). Furthermore, compared with Nor-CM, Hypo-CM treatment led to fewer total cells and neutrophils in BALF, lower MPO activity in the lung tissue, and more significantly reduced levels of pro-inflammatory cytokines in the models.

### 2.5. Hypo-CM Ameliorated Lung Inflammation through Modulating Polarization of Anti-Inflammatory Macrophages

To assess whether the pro-resolving effect of Hypo-CM in ALI models correlated with the polarization of anti-inflammatory macrophages, we investigated phenotypes of alveolar macrophages (AMs) in BALF of mice. The adherent cells from BALF were collected, and their purity was analyzed. As shown in Figure 6A, these cells were identified as CD11b^−^CD45^+^siglec-F^+^CD11c^+^, which was consistent with the markers of AMs [41]. MSCs-CM treatment increased the level of CD206^+^ AMs, whereas decreasing the level of CD80^+^ AMs in ALI mice. Moreover, compared with Nor-CM, Hypo-CM treatment induced a significantly higher level of CD206^+^ and lower level of CD80^+^ AMs in the alveolus (Figure 6B–D), suggesting that Hypo-CM treatment resulted in more anti-inflammatory AMs than Nor-CM.

To further validate the advantageous effect of Hypo-CM over Nor-CM in modulating polarization of anti-inflammatory macrophages, we detected the expression of Arginase-1 (Arg-1) and inducible nitric oxide synthase (iNOS) in the lung tissues. We found that the number of F4/80^+^ Arg-1^+^ lung macrophages, indicative of anti-inflammatory macrophages, significantly decreased in ALI mice, and MSCs-CM treatment increased the number of F4/80^+^ Arg-1^+^ lung macrophages. Compared with Nor-CM, the effect of Hypo-CM was more obvious (Figure 7A,B). On the contrary, compared with ALI mice, mice treated with MSCs-CM exhibited a lower number of F4/80^+^ iNOS^+^ macrophages in the lungs, particularly in the lungs of mice treated with Hypo-CM (Figure 7C,D). Therefore, these results suggested that Hypo-CM was more efficient than Nor-CM in promoting polarization of anti-inflammatory macrophages in ALI mice.

### 2.6. Pro-Resolving Effect of Hypo-CM Was Associated with Efferocytosis of Anti-Inflammatory Macrophages

Efferocytosis of macrophages is an important process involved in polarization of anti-inflammatory macrophages and resolution of inflammation [42]. In order to elucidate the mechanism of Hypo-CM in limiting the inflammation, we detected the efferocytic activity of lung macrophages in lung tissues. We first determined the abundance of apoptotic cells (ACs) at 48 h post injection of LPS in animal models. As shown in Figure 8A, few ACs were present in control mice injected with PBS alone, but much more ACs accumulated in the lungs of ALI mice. However, the number of ACs reduced significantly upon administration with either Nor-CM or Hypo-CM, especially with Hypo-CM (Figure 8A,B). Conversely, CD206^+^ lung macrophages were scarce in ALI mice, whereas Nor-CM and Hypo-CM significantly increased CD206^+^ lung macrophages (Figure 8A,C), which was consistent with the results of flow cytometry in Figure 6D. Moreover, the ratio of free ACs to CD206^+^ macrophages-associated ACs, indicative of defective efferocytosis of macrophages, was highest in lung tissues of ALI models. In ALI mice infused with MSCs-CM, the ratio of free ACs number to macrophages-associated ACs number was downregulated. Compared with Nor-CM, the effect of Hypo-CM was more potent (Figure 8A,D), suggesting an advantageous effect of Hypo-CM over Nor-CM in restoring efferocytic activity of lung macrophages in ALI models.

## 3. Discussion

Oxygen is indispensable for tissue cells because of its key role in energy metabolism. Given that oxygen concentration in stem cells niches in vivo is much lower than 21%, which is traditionally used in the cell culture systems, hypoxia treatment has been used to modulate various functions of MSCs. In general, two modes of hypoxia treatment were used: short-term hypoxia treatment (or hypoxia preconditioning), and long-term hypoxia treatment (or consecutive hypoxia). Several studies have demonstrated that low oxygen tensions in the culture system help to maintain the undifferentiated state of stem cells, increase expression of pluripotency markers, and preserve cell function [36,43]. However, the effects of consecutive hypoxic culture on the proliferation ability of MSCs remain controversial. For instance, Jin et al., demonstrated that consecutive hypoxic culture enhanced the proliferation of human bone marrow-derived MSCs (BMSCs) [37]. Similar results were shown in another study on human MSCs derived from umbilical cord blood [43]. Conversely, Pezzi et al., reported that such consecutive hypoxia inhibited the proliferation of human BMSCs with decreased mitochondrial activity [27]. Our previous study also showed that the proliferation rate of rat BMSCs decreased when cells were consecutively cultured in 1% O_2_. In this study, we showed that consecutive hypoxic culture not only greatly enhanced the proliferation ability, but also attenuated senescence of UC-MSCs. It suggests that consecutive hypoxic cultures can serve as a strategy to promote the production of UC-MSCs with optimized quality. Moreover, the decrease in the secretion of SASP factors by Hypo-MSCs, such as IL6 and IL8, is beneficial to the immunomodulatory function of Hypo-CM. Further studies are needed to clarify the effects of different oxygen concentrations on the proliferation and quality of UC-MSCs.

The immunomodulatory effects of MSCs mainly owe to the soluble components or factors secreted by these cells [44,45,46]. In preclinical studies and clinical application, secretome or conditioned medium of MSCs have several advantages over MSCs. Studies showed that only a small amount of MSCs remained alive 24–72 h after being infused in vivo [14]. Conditioned medium is more convenient in availability and storage. In this study, both Nor-CM and Hypo-CM could effectively attenuate lung inflammation in ALI models. Our previous study also demonstrated that conditioned medium from rat BMSCs could mitigate ischemia/reperfusion-induced cerebral injury [23]. These data suggested that secretome or conditioned medium could be an ideal choice of MSCs-based products in clinical applications.

Macrophages and neutrophils are dominant immune cells mediating immune response in the early stage of ALI. Infiltration and activation of pro-inflammatory macrophages and neutrophils help to clear the pathogens. However, excessive activation of these cells leads to systematic inflammatory reaction syndrome and multiple organ dysfunction, and even death of the patients. Factors in promoting activation of anti-inflammatory macrophages, such as TSG6, IL4, IL13, IL10, and glucocorticoids, can mitigate the inflammation and promote tissue repair [38]. Our results showed that compared with Nor-CM, Hypo-CM was more efficient in modulating the polarization of anti-inflammatory macrophages and limiting lung inflammation in ALI mice. In agreement with our findings, a previous study showed that MSCs-CM collected from long-term hypoxic cultures (10% O_2_) is also more effective than that from normoxic cultures in limiting cerebral inflammation of experimental autoimmune encephalomyelitis. Moreover, UC-MSCs obtained from this hypoxic culture system could secret higher levels of insulin-like growth factor 2, which contributed to the enhanced therapeutic effects of MSCs-CM [11]. In this long-term hypoxic culture system, which was different from ours, UC-MSCs were cultured first for several passages under normoxic conditions and then cultured in hypoxia for another three passages [11]. Whether these two modes of long-term hypoxic culture make a difference in regulating immunomodulatory function, as well as proliferation, senescence, and other biological properties of UC-MSCs, needs to be further investigated.

Efferocytosis is a process in which phagocytes engulf apoptotic and dead cells to prevent tissue necrosis and inflammation [3,47]. The process of efferocytosis promotes polarization of anti-inflammatory macrophages. Defects in efferocytic activity of macrophages were associated with persistence and chronicity of lung inflammation, such as chronic obstructive pulmonary disease [48]. Anti-inflammatory or pro-resolving macrophages are key cells to execute efferocytosis during the repair stage of injured tissues [40,49,50]. Studies showed that efferocytosis of macrophages in ARDS patients was impaired due to an increase in polarization of pro-inflammatory AMs. Approaches that improved efferocytosis of macrophages helped to alleviate ARDS/ALI [8,51]. An observation on MSCs therapy revealed that infusion of MSCs promoted clearance of the recruited neutrophils and resolution of systemic inflammation in repeated social defeat mice models [52]. In this study, we showed that MSCs-CM promoted the polarization of anti-inflammatory lung macrophages, and this effect was closely associated with enhanced efferocytosis in the lung tissues of ALI models. More importantly, both in vitro and in vivo studies verified that Hypo-CM was more effective in modulating efferocytosis of lung macrophages than Nor-CM. Our study also demonstrated that Hypo-MSCs produced significantly higher levels of immunomodulatory factors, including PGE2, IGF1, IL10, TSG6, and TGFβ. These paracrine factors serve as key contributors of MSCs-mediated immunomodulation of macrophages, which enable Hypo-CM to exhibit significant advantages over Nor-CM in limiting lung inflammation. PGE2 can promote polarization of anti-inflammatory macrophages [53] and efferocytosis [54]. IGF-1 has been demonstrated to reduce inflammation levels in chronic experimental Chagas disease [55]. IL10 could enable macrophages to internalize apoptotic cells by affecting the guanine nucleotide exchange factor Vav1 in macrophages [42] and enhance efferocytosis of macrophages through promoting CD206^+^ anti-inflammatory macrophages [56]. TSG6 can also promote polarization of anti-inflammatory macrophages and promote resolution of lung inflammation [57]. For instance, knockdown of TSG6 mitigated the effects of MSCs in limiting lung inflammation [58]. As for TGFβ, in addition to its anti-inflammatory function, it can also promote conversion of resident fibroblasts into active myofibroblasts when overactivated, and induce lung fibrosis. However, immunomodulatory factors such as TSG6 and PGE2 in MSCs-CM are anti-fibrotic, which can antagonize or minimize the potential pro-fibrotic effects of TGFβ [59,60]. Other cytokines, such as keratinocyte growth factor (KGF) [46], angiopoietin 1 (ANGPT1, also known as Ang-1) [61], and hepatocyte growth factor (HGF) [62,63] can also mediate the therapeutic effects of MSCs in alleviating inflammation of ALI.

In addition to these cytokines, exosomes and extracellular vesicles may be potential contributors as well. Moreover, the therapeutic effects of MSCs in ALI/ARDS are multifaceted. In addition to anti-inflammation, they have abilities toward anti-oxidative stress, anti-fibrosis, pro-angiogenesis, and anti-apoptosis, by acting on epithelial cells [64], endothelial cells [65], and T cells [66], etc., in pulmonary tissues. Further studies are needed to clarify the effects of Hypo-CM on these cells as well as the effective molecules mediating the beneficial effects of Hypo-CM in our system.

## 4. Materials and Methods

### 4.1. Experimental Animals

Eight-week-old C57BL/6 mice (18–20 g) were purchased from Sanxia University Animal Center. The animals were housed in a specific pathogen-free condition with controlled temperature and a 12 h light/dark cycle. All animal experimental protocols were conducted by the Local Ethical Committee on Animal Care and Use of Wuhan University, Wuhan, China (permit number: WQ20210229, 3 March 2021).

### 4.2. Culture of UC-MSCs

Human umbilical cords were derived from healthy term (gestational age ≥ 37 weeks) deliveries under informed consent and were approved by the Medical Ethic Committee of Wuhan University School of Medicine (permit number: JC2020-008, 28 March 2020). UC-MSCs were isolated as previously described [11]. Briefly, the umbilical cord was rinsed with PBS and dissected into small pieces. The pieces were placed in culture flasks and cultured in low-glucose Dulbecco’s Modified Eagle Medium (DMEM, Gibco, Big Cabin, OK, USA) containing 10% (vol/vol) fetal bovine serum (FBS, ScienCell Research Laboratories, Carlsbad, CA, USA) and 1% (vol/vol) penicillin–streptomycin. They were incubated respectively under normoxic conditions (5% CO_2,_ 21% O_2_) or hypoxic conditions (94% N_2_, 5% CO_2_, 1% O_2_) at 37 °C in the humidified atmosphere. Cells were recorded as passage 0. The medium was replenished every 3–4 days. Upon reaching 80% confluence, cells were trypsinized and subcultured in normoxic conditions or hypoxic conditions as in passage 0. Cells of passage 2 to passage 6 were used in this study.

### 4.3. CCK8 Assay

For analysis of the proliferation rate, UC-MSCs (passage 3 or passage 4) from normoxic cultures, hypoxic cultures were respectively seeded into 96-well plates at a density of 2 × 10^3^ cells per well. They were cultured in normoxic or hypoxic condition. For hypoxia preconditioning, MSCs from normoxic cultures were seeded in 96-well plates and then cultured hypoxic condition. The proliferation rate of UC-MSCs was then detected in 0, 24, and 48 h by Cell Counting Kit-8 (CCK8) (Meilunbio, Dalian, China) according to the manufacturer’s instructions.

### 4.4. Senescence-Associated β-Galactosidase (SA-β-gal) Staining

SA-β-gal assay was performed according to the manufacturer’s instructions (Beyotime, C0602, Nantong, China). Briefly, cells were fixed in PBS containing 1% formaldehyde and 0.2% glutaraldehyde for 10 min at RT. After twice washing with PBS, cells were incubated in β-galactosidase staining solution containing 5% x-gal overnight at 37 °C. Cells were then washed with PBS and images were captured under microscope (Olympus, Hamburg, Germany). SA-β-gal positive cells were counted using Image J software.

### 4.5. Preparation of MSCs-CM

UC-MSCs from normoxic and hypoxic cultures were respectively seeded at a density of 1 × 10^6^ per 75 cm^2^ culture flask in complete DMEM medium for 24 h. The medium was replaced by serum-free DMEM and then incubated in the normoxic or hypoxic condition for another 48 h. For hypoxia preconditioning cultures, the MSCs from normoxic cultures were incubated in hypoxic condition for another 48 h. The culture medium was then collected, and cell numbers were determined by cell counter. The culture medium was filtered using a 0.22 μm filter to remove the cellular debris. Supernatant concentration was adjusted to 1 × 10^5^ mL according to the harvested cell numbers. The supernatant of UC-MSCs from normoxic cultures was termed Nor-CM, and supernatant from hypoxic cultures was termed Hypo-CM. For in vivo animal experiments, the supernatant was concentrated about 200-fold by centrifugation in a 3-KD ultrafiltration tube (Millipore, UFC900396, Darmstadt, Germany) and stored at −80 °C.

### 4.6. Culture of BMDMs

Culture of mice BMDMs was performed as previously described [67]. Briefly, both femurs and tibias of mice were collected and washed in sterile PBS. Both ends of femurs and tibias were removed, and bone marrow cavities were flushed with RPMI 1640 medium (Gibco, Big Cabin, OK, USA). Cell suspension was centrifuged at 200× *g* for 6 min at room temperature (RT). Afterward, cells were collected and cultured in RPMI 1640 medium containing 10% FBS, 20% supernatants of L929 mouse fibroblasts as conditioned medium providing macrophage colony-stimulating factor (M-CSF). The medium was replenished every 2–3 days. Cells were collected as BMDMs on day 6, stained with antibodies of CD11b (Miltenyi, 130113806, Bergisch Gladbach, Germany) and F4/80 (Miltenyi, 130117509, Bergisch Gladbach, Germany), and analyzed through flow cytometry (BD, Franklin Lakes, NJ, USA).

For induction of polarization, BMDMs (1 × 10^6^) were stimulated with either 1 μg/mL LPS or 20 ng/mL IL-4 for 48 h in the presence or absence of MSCs-CM (from 2 × 10^5^ MSCs). BMDMs were then collected, stained with antibodies of CD206 (Biolegend, cat no: 141712, San Diego, CA, USA) and CD80 (Miltenyi, cat no: 130102883, Bergisch Gladbach, Germany), and analyzed through flow cytometry.

### 4.7. ALI Induction and MSCs-CM Treatment

To induce ALI, mice were anesthetized. A single dose of 50 μL LPS (5 mg/kg body weight, Sigma-Aldrich, 0111: B4, St. Louis, MO, USA) was administered to the mice via noninvasive intracheal instillation (i.t.). An equal volume of phosphate-buffered saline (PBS) was administered in the same way to control mice. Four hours later, mice were given 50 μL MSCs-CM (from 1 × 10^6^ MSCs) via i.t. for treatment. The same volume of PBS was similarly administered to the control mice or ALI model mice. Animals were anesthetized and sacrificed at the end of 48 h (Figure 9).

### 4.8. Isolation and Identification of AMs

Mice AMs were obtained as previously described [68], and their phenotype was analyzed. Briefly, bronchoalveolar lavage fluid (BALF) from C57BL/6 mice was collected by inserting a 22G needle cannula into trachea. Ice-cold PBS was slowly instilled into the lung, and then the lavage fluid was collected as BALF. The BALF was centrifuged and cell pellets were collected. Cells were cultured in RPMI 1640 complete medium for 6 h. Then, adherent cells were collected as AMs. For identification of AMs, cells were stained with antibodies of CD11c (Miltenyi, cat no: 130122939, Bergisch Gladbach, Germany), CD11b (Miltenyi, cat no: 130113806, Bergisch Gladbach, Germany), CD45 (BD, cat no: 561018, Franklin Lakes, NJ, USA) and siglec-F (Miltenyi, cat no: 130123816, Bergisch Gladbach, Germany) and analyzed by flow cytometry. For detection of CD206 and CD80 expression in AMs, these cells were stained with antibodies of CD206 (Biolegend, cat no: 141712, San Diego, CA, USA) and CD80 (Miltenyi, cat no: 130102883, Bergisch Gladbach, Germany), and analyzed through flow cytometry.

### 4.9. Analysis of Alveolar Neutrophils

BALF cells were collected as described above, and neutrophils were analyzed by staining these cells with antibodies of CD45 (BD, cat no: 561018, Franklin Lakes, NJ, USA), CD11b (Miltenyi, cat no: 130113806, Bergisch Gladbach, Germany) and Gr-1 (Miltenyi, cat no: 130102837, Bergisch Gladbach, Germany). Results were analyzed through flow cytometry.

### 4.10. MPO Assay

MPO activity was conducted by commercial MPO assay kit according to the manufacturer’s instructions (Nanjing Jiancheng, Bioengineering Institute, A044-1-1, Nanjing, China). Briefly, fresh lung tissues were isolated from mice, weighed, and homogenized. Then, supernatant was collected and MPO reaction was performed as following: 200 μL samples were added into 3.25 mL of reaction buffer for 10 min at 60 °C. The absorption of samples was measured at 460 nm by spectrophotometer (SpectraMax, San Jose, CA, USA).

### 4.11. Enzyme-Linked Immunosorbent Assay (ELISA)

Levels of cytokines in supernatant of human UC-MSCs or in serum were determined by commercially available ELISA kits (Bioswamp, Wuhan, China).

### 4.12. Analysis of Cell Apoptosis

Cells were harvested and washed three times with PBS. Then, cells were stained with Annexin-V and propidium iodide (PI) according to the manufacturer’s instructions. Briefly, cells were suspended in binding buffer, mixed and then incubated with Annexin-V for 15 min at 4 °C. PI was added into cell suspensions, and the samples were analyzed for 5 min by flow cytometry.

### 4.13. Induction of Apoptosis and Fluorescent Labeling of Jurkat Cells

Jurkat cells were induced to apoptosis by treating with 1 μM staurosporine (Meilunbio, 62996741, Dalian, China) in serum-free RPMI 1640 medium and incubated at 37 °C, 5% CO_2_ for 20 h. Cells were collected, labeled with Annexin-V and PI, and cell apoptosis was analyzed through flow cytometry. The apoptotic cells were washed twice with PBS and resuspended in 1 mL Dilent C (Maokangbio, MX4021, Shanghai, China) at a density of 2 × 10^7^/mL. The Dilent C containing apoptotic Jurkat cells was then mixed with 1 mL 2 μM PKH26 dye and incubated for 5 min. After washing twice with RPMI 1640, the cells were then used for the following experiments.

### 4.14. In Vitro Efferocytosis Assay

BMDMs were seeded in a 6-well plate at a density of 1 × 10^6^ per well and cultured with or without MSCs-CM for 48 h in the presence of LPS (1 μg/mL). BMDMs were adherent cells, whereas Jurkat cells were non-adherent. BMDMs were co-cultured with 5 × 10^6^ PKH26-labeled apoptotic Jurkat cells for 45 min, followed by washing twice with PBS to remove the cells in suspension or unengulfed Jurkat cells. BMDMs were fixed with 4% formaldehyde for 10 min and then washed twice with PBS. The experimental schema was shown in Figure 10. Images were captured by the fluorescence microscope (Olympus, Hamburg, Germany).

### 4.15. Immunofluorescence Microscopy

For analyzing the phenotype of macrophages in the pulmonary tissues, lung tissues were collected and fixed with formalin. After paraffin-embedded and sectioned, the lung specimens were stained with antibodies of F4/80 (Abcam, cat no: ab6640, Boston, MA, USA), iNOS (Abcam, cat no: ab283655, Boston, MA, USA) and Arg-1 (CST, cat no: 93668, Danvers, MA, USA). Cell nuclei were counterstained with DAPI (Beyotime, cat no: C1003, Nantong, China). Images were captured by laser scanning confocal microscopy (Leica, 63×objective, Wetzlar, Germany).

For in vivo assay of lung efferocytosis, the sectioned lung specimens were incubated with TUNEL reagents (Beyotime, cat no: C1091, Nantong, China) and stained with antibodies of CD206 (CST, cat no: 24595, Danvers, MA, USA). Cell nuclei were counterstained with DAPI (Beyotime, cat no: C1003, Nantong, China). Images were captured by laser scanning confocal microscopy (Leica, Wetzlar, Germany). Analysis of efferocytosis of macrophages was described previously [69]. Briefly, the level of efferocytosis in the pulmonary tissues was quantified by counting TUNEL^+^ nuclei that were associated with macrophages (“associated”). Macrophage-associated apoptotic cells were cells with TUNEL^+^ nuclei surrounded by or in contact with neighboring CD206^+^ macrophages. Free apoptotic cells were not in contact with neighboring CD206^+^ macrophages.

### 4.16. Statistical Analysis

Data were presented as mean ± SD and analyzed by GraphPad Prism 5 software (GraphPad Software Inc., San Diego, CA, USA). For comparisons of two groups, two-tailed unpaired Student’s *t* test was conducted. One-way ANOVA was applied to compare more than two groups. *p* value of less than 0.05 was considered as statistically significant.

## 5. Conclusions

In conclusion, our findings provide direct evidence that: (1) conditioned medium from UC-MSCs can promote polarization of anti-inflammatory macrophages with increased capacity in efferocytosis and can effectively promote resolution of lung inflammation in ALI models; (2) consecutive hypoxic culture can not only enhance the proliferation of UC-MSCs with good cell quality but can also improve immunomodulatory and therapeutic efficacy of the secretome of UC-MSCs. Thus, consecutive hypoxic culture can serve as a strategy to produce UC-MSCs and the secretome with optimized immunomodulatory properties for therapy of inflammatory disorders such as ALI.

## Figures and Tables

**Figure 1 ijms-23-04333-f001:**
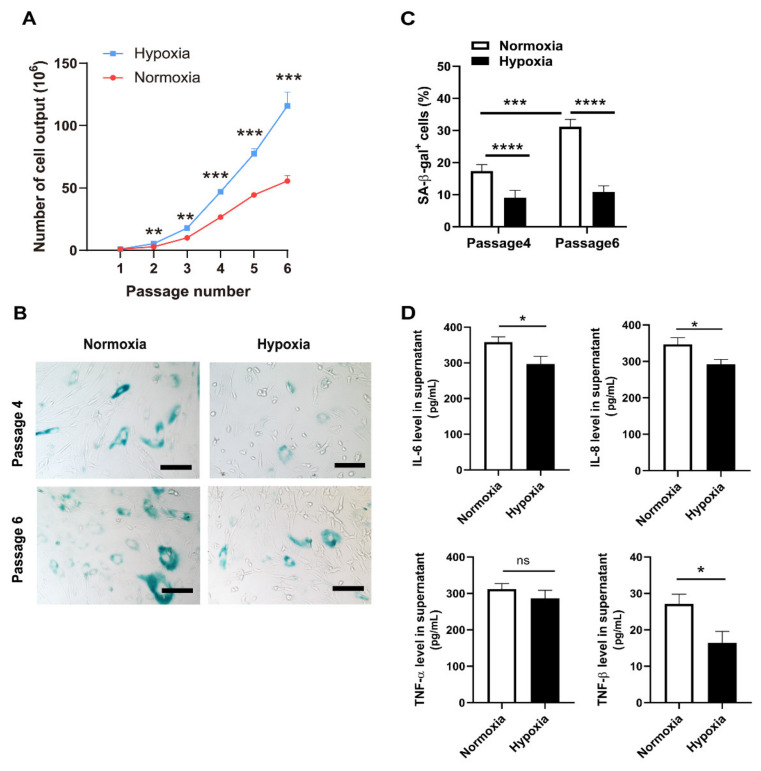
Hypoxic culture-enhanced proliferation and inhibited senescence of MSCs. (**A**) Quantification of cell output from passage 1 to passage 6 cultured in normoxic and hypoxic condition. (**B,C**) Analysis of SA-β-gal activity (SA-β-gal^+^ cells) in UC-MSCs of passage 4 and passage 6. SA-β-gal^+^ cells were stained in light blue (**B**) and quantified (**C**); scale bar = 100 μm. (**D**) Analysis of SASP cytokines (IL6, IL8, TNFα, and TNFβ) in the supernatants from UC-MSCs through ELISA. Cells of passage 3 or passage 4 were seeded in 6-well plates, and supernatants were collected 48 h later. * *p* < 0.05, ** *p* < 0.01, *** *p* < 0.001 and **** *p* < 0.0001. UC-MSCs, umbilical cord-derived mesenchymal stem cells; SA-β-gal, senescence-associated β-galactosidase; ELISA, enzyme-linked immunosorbent assay; ns, no statistical significance; SASP, senescence-associated secretory phenotype.

**Figure 2 ijms-23-04333-f002:**
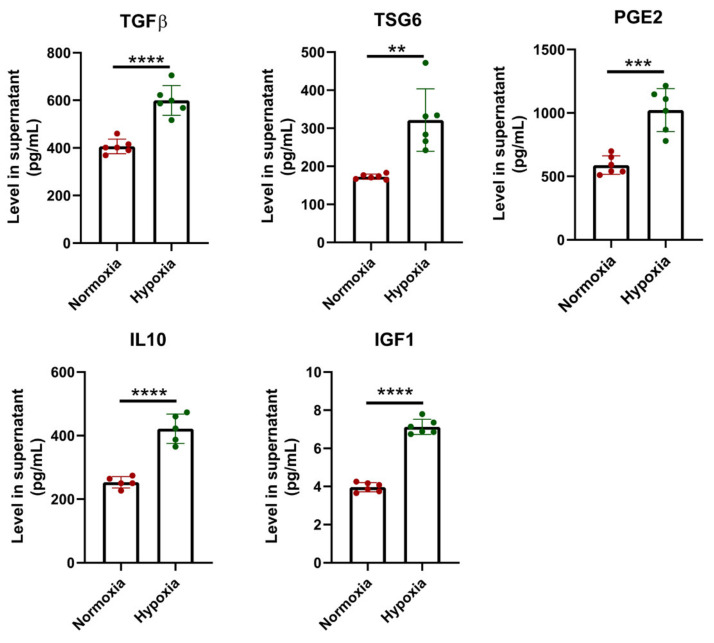
Consecutive hypoxia promoted UC-MSCs to secret high levels of immunomodulatory factors. The level of immunomodulatory factors (TGFβ, TSG6, PGE2, IL10, and IGF1) in the conditioned medium from UC-MSCs was analyzed through ELISA. Cells of passage 3 or passage 4 were seeded in 6-well plates, and supernatants were collected 48 h later. Each dot represents data for one biological sample. ** *p* < 0.01, *** *p* < 0.001 and **** *p* < 0.0001. TSG6, TNFα-stimulated gene 6; PGE2, prostaglandin E2; IGF1; insulin-like growth factor 1.

**Figure 3 ijms-23-04333-f003:**
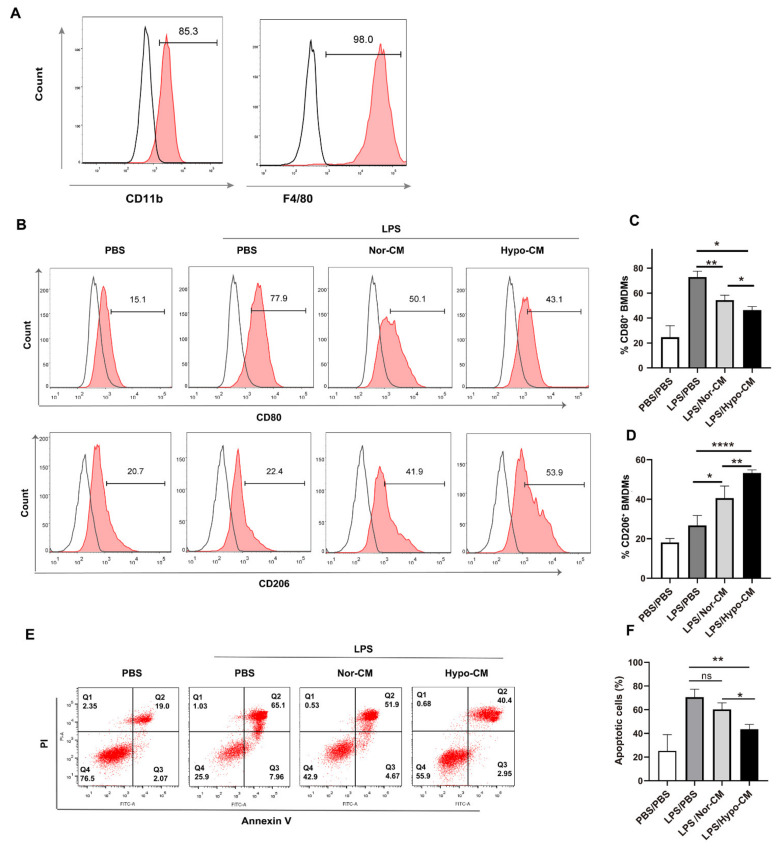
Hypo-CM promoted anti-inflammatory polarization of macrophages in vitro. (**A**) Characterization of mice BMDMs through flow cytometry. (**B**–**D**) Phenotype of BMDMs upon stimulation with LPS (1 μg/mL) for 48 h in the presence of Nor-CM or Hypo-CM. (**E**,**F**) Analysis of apoptosis level of BMDMs. BMDMs were stimulated by LPS (1 μg/mL) in the presence of Nor-CM or Hypo-CM for 48 h, stained with Annexin V/PI, and analyzed through flow cytometry. The data are representative of three independent experiments. * *p* < 0.05, ** *p* < 0.01 and **** *p* < 0.0001. Statistics for multiple comparisons were analyzed with one-way ANOVA. BMDMs, bone marrow-derived macrophages; LPS, lipopolysaccharide; PI, propidium iodide; PBS, phosphate-buffered saline; ns, no statistical significance.

**Figure 4 ijms-23-04333-f004:**
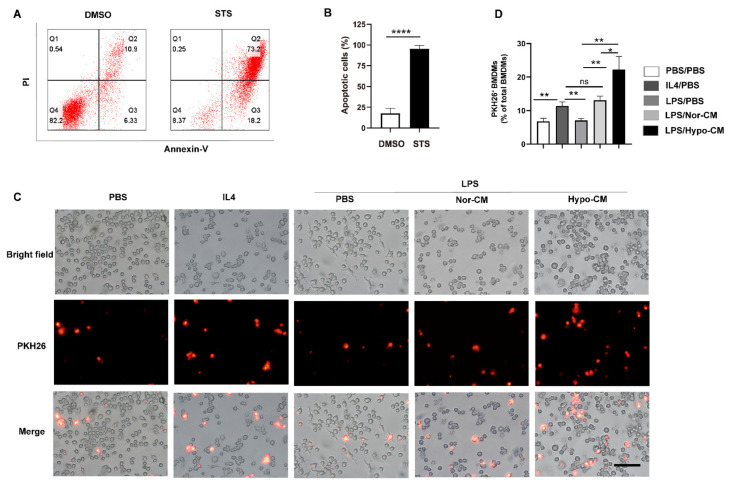
Hypo-CM enhanced efferocytosis of macrophages in vitro. (**A**,**B**) Detection of apoptosis level of Jurkat cells. Apoptosis of cells was induced by STS (1 μM) for 20 h. Then, cells were stained with AnnexinV/PI and analyzed by flow cytometry. DMSO was used as blank control. (**C**,**D**) Analysis of efferocytosis in BMDMs. Mice BMDMs (1 × 10^6^) were stimulated with either LPS (1 μg/mL) or IL-4 (20 ng/mL), and cultured with or without the presence of Nor-CM or Hypo-CM for 48 h. They were then co-cultured with PKH26-labeled apoptotic Jurkat cells (5 × 10^6^) for 45 min and analyzed through fluorescence microscopy. Representative images of BMDMs engulfing apoptotic Jurkat (red) are shown in (**C**). Scale bar = 100 μm. The ratio of BMDMs engulfing apoptotic cells to all BMDMs is quantified in (**D**). The data are representative of three independent experiments. * *p* < 0.05, ** *p* < 0.01, **** *p* < 0.0001. STS, staurosporine; ns, no statistical significance.

**Figure 5 ijms-23-04333-f005:**
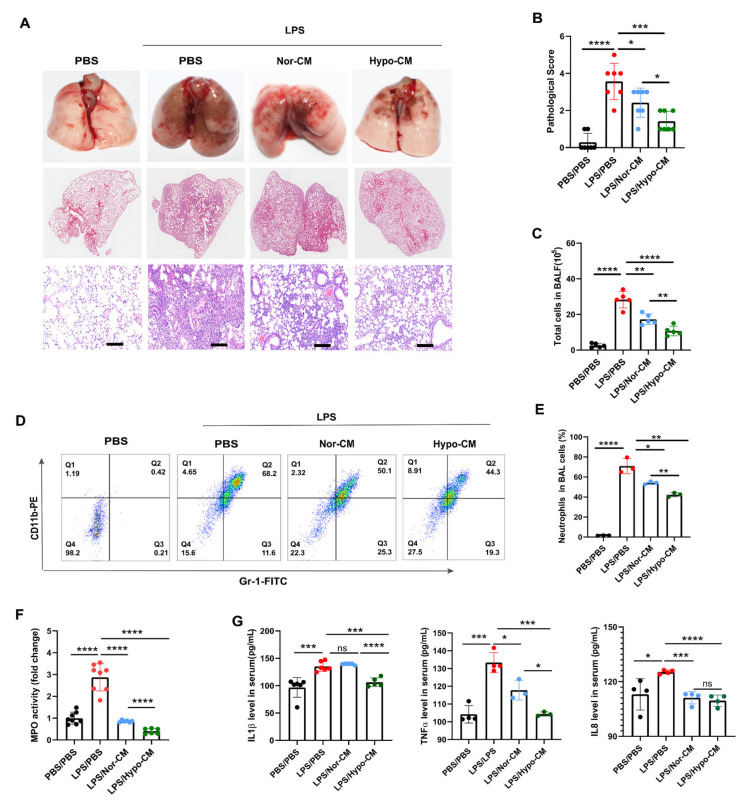
Hypo-CM effectively promoted resolution of inflammation in the lungs of ALI models. (**A**) Representative images of lung tissues and H&E staining of the lungs. Scale bar: 200 μm. (**B**) Quantification of lung injury through pathological score evaluation, *n* = 7. (**C**) Quantification of total cells in BALF. Cells in BALF were harvested 48 h after LPS injection and then counted. *n* = 5. (**D**) Analysis of neutrophils (CD11b^+^Gr-1^+^) level in BALF through flow cytometry. *n* = 3. (**E**) Quantification of the proportion of alveolar neutrophils in total BALF cells. (**F**) Analysis of MPO activity in the lungs. Fresh mice lung tissues were collected 48 h after LPS administration and used for measurement of MPO activity. *n* = 6~8. (**G**) Analysis of inflammatory factors in the serum through ELISA. Each dot represents data from one animal. *n* = 3~6. * *p* < 0.05, ** *p* < 0.01, *** *p* < 0.001 and **** *p* < 0.0001. Statistics for multiple comparisons were obtained from one-way ANOVA. ALI, acute lung injury; i.t., intracheal instillation; MPO, myeloperoxidase; BALF, bronchoalveolar lavage fluid; ns, no statistical significance.

**Figure 6 ijms-23-04333-f006:**
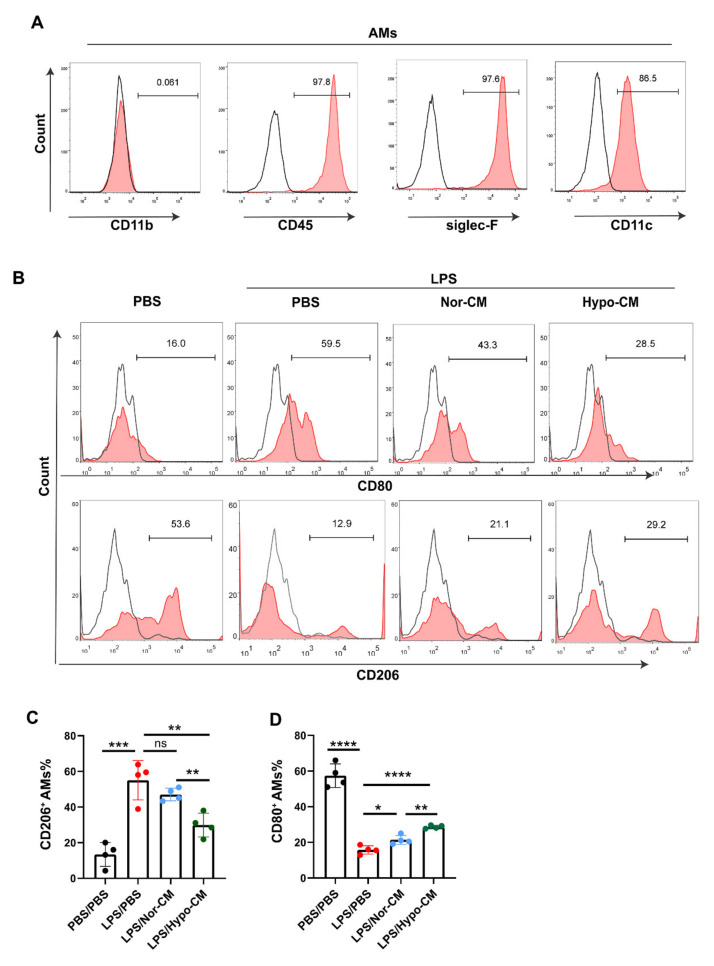
Hypo-CM promoted polarization of anti-inflammatory AMs in ALI models. (**A**) Phenotypic characterization of AMs through flow cytometry. (**B**–**D**) Analysis of CD80 and CD206 expression in AMs. BALF was collected from mice 48 h post LPS administration, and AMs were then isolated and analyzed through flow cytometry. *n* = 4. Each dot represents data from one animal. * *p* < 0.05, ** *p* < 0.01, *** *p* < 0.001 and **** *p* < 0.0001. Statistics for multiple comparisons were obtained from one-way ANOVA. AMs, alveolar macrophages; ns, no statistical significance.

**Figure 7 ijms-23-04333-f007:**
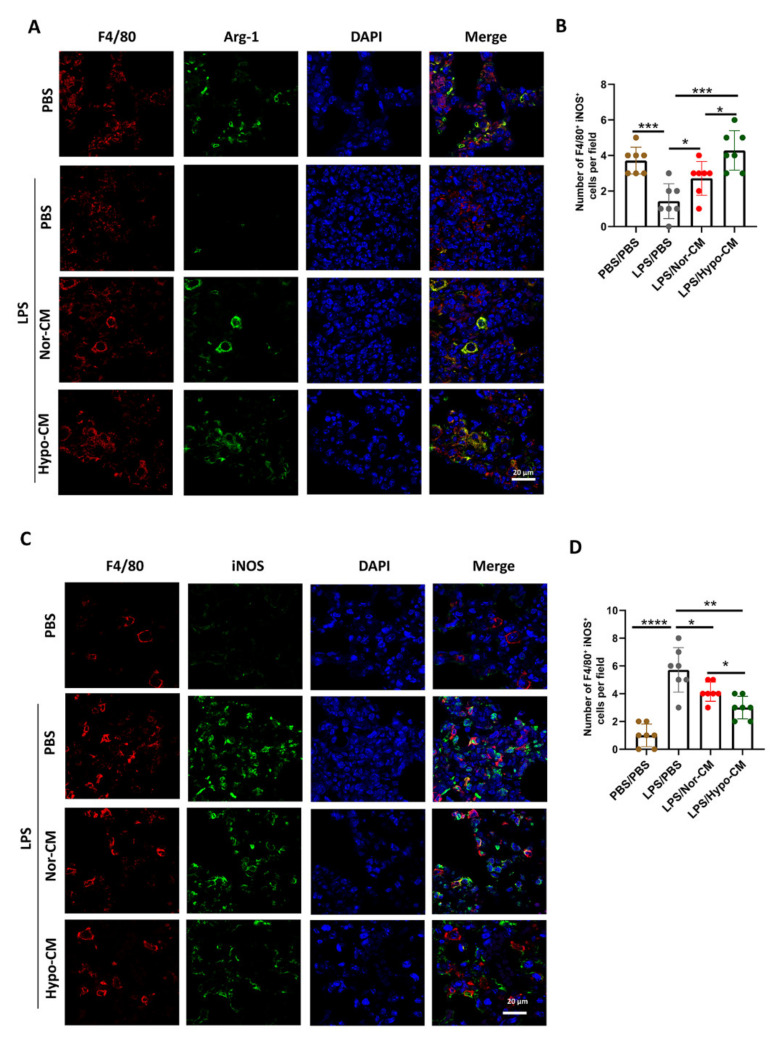
Hypo-CM promoted polarization of anti-inflammatory lung macrophages in lung tissues. (**A**) Representative images of colocalization of F4/80^+^ (red) and Arg-1^+^ (green) in lung tissues. Cell nuclei were stained with DAPI (blue). Scale bar = 20 μm. *n* = 7. (**B**) Quantification of the positive cells of F4/80^+^ Arg-1^+^ in the lung sections from mice in (**A**). (**C**) Representative images of colocalization of F4/80^+^ (red) and iNOS^+^ (green) in lung tissues. Cell nuclei were stained with DAPI (blue). Scale bar = 20 μm. *n* = 7. (**D**) Quantification of the positive cells of F4/80^+^iNOS ^+^ in the lung tissues from mice in (**C**). * *p* < 0.05, ** *p* < 0.01, *** *p* < 0.001 and **** *p* < 0.0001. Arg-1, Arginase-1; iNOS, inducible nitric oxide synthase; DAPI, 4′,6-diamidino-2-phenylindole.

**Figure 8 ijms-23-04333-f008:**
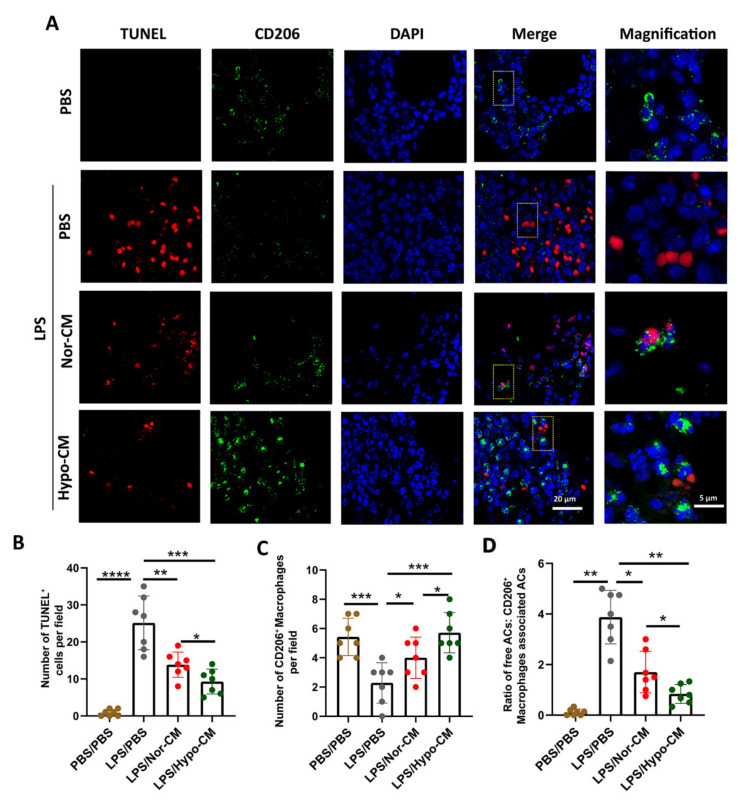
Pro-resolving effect of Hypo-CM in ALI was associated with the efferocytosis of anti-inflammatory macrophages. (**A**) Representative images of ACs (TUNEL^+^, red) and CD206^+^ macrophages (green) in lung tissues. Cell nuclei were stained with DAPI (blue). Scale bar = 20 μm. Yellow rectangles indicated the presence or absence of engulfed TUNEL^+^ red fluorescence signals, which were magnified in the far-right panel (Scale bar = 5 μm). *n* = 7. (**B**) Quantification of the ACs (TUNEL^+^, red) in lung sections from mice in (**A**). (**C**) Quantification of the CD206^+^ macrophages (green) in lung sections from mice in (**A**). (**D**) Quantification of the ratio of free ACs: macrophage-associated ACs from mice in (**A**). Macrophage-associated apoptotic cells followed the criteria of TUNEL^+^ nuclei surrounded by or in contact with neighboring CD206^+^ macrophages. Free apoptotic cells exhibited nuclear condensation, were not in contact with neighboring CD206^+^ macrophages. The ratio of free ACs:macrophage-associated ACs was indicative of the defective efferocytosis of macrophages. * *p* < 0.05, ** *p* < 0.01 *** *p* < 0.001 and **** *p* < 0.0001. ACs, apoptotic cells; TUNEL, terminal deoxynucleotidyl transferase-mediated dUTP nick end labeling.

**Figure 9 ijms-23-04333-f009:**
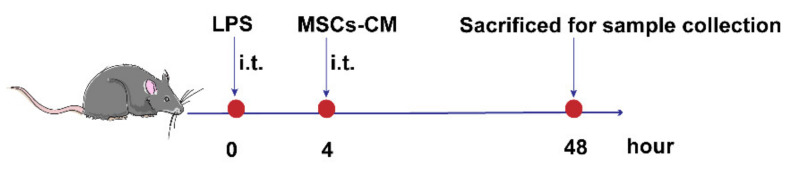
Experimental schema of in vivo study on ALI models treated with Nor-CM or Hypo-CM.

**Figure 10 ijms-23-04333-f010:**
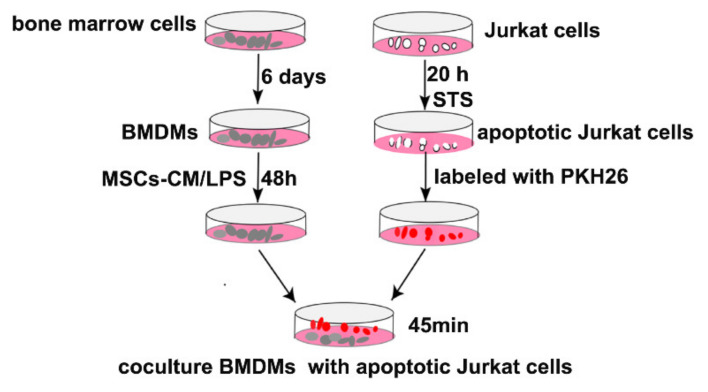
Schematic diagram of the methodology of efferocytosis assay in vitro.

## Data Availability

The datasets used and/or analyzed during the current study are available from the corresponding author on reasonable request.

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
