# Peer review of "Secretome of Mesenchymal Stem Cells from Consecutive Hypoxic Cultures Promotes Resolution of Lung Inflammation by Reprogramming Anti-Inflammatory Macrophages"

_ijms, 2022, doi:10.3390/ijms23084333_

Round 1

Reviewer 1 Report

The present study addressed to an actual issue, more precisely investigate how the secretion of mesenchymal stem cells (MSCs) influences the polarization of the lung macrophages and the effect of the conditioned media of MSCs (CM-MSC) in reducing pathogenic phenomena in an experimental animal model of acute lung injury (ALI). The MSCs studied are human MSCs isolated from the umbilical cord (UC-MSC).

All in all, the study hypothesis is interesting, addresses an actual research field and is based on relevant bibliographic data. The methodology and the experimental procedures are adequate for the aims of the study. However, some aspects presented in the paper need clarification, as follows:

  • The working methodology is mainly related to the objectives, but the study includes many different experimental procedures (in vitro, in vivo, ex vivo) in which effects and/or responses of either human or murine cells are studied, and their exposure can be easily confused for the reader.
  • The experimental set-up represented as a diagram is included in the results chapter (figure 4A, figure 5A) although its place would be in the methods chapter.
  • The supplementary material show the results obtained by the CCK8 assay, but the working method is not described in the paper.
  • Some experimental data were obtained using ”cells in passage 3-4” (Figure 1, Figure 2). How is this passage defined? Are the cells after the third passage? Are there cells from two consecutive passages, P3 and P4?
  • It seems that figures 1C and 1D show the same results, so there is an unnecessary overlap.
  • In the results, subchapter 2.3., it is said that "Hypo-MSCs secreted higher levels of anti-inflammatory cytokines and lower levels of pro-inflammatory cytokines." Indeed, the previous subchapter (2.2.) demonstrates that MSCs under consecutive hypoxia secrete a number of anti-inflammatory cytokines, but there are no data presented in the paper related with the levels of proinflammatory cytokines secreted by these cells.
  • Figure 3A, characterization of mice BMDMs through flow cytometry, does not indicate the cell type used as a negative control.
  • In Figures 3C and 3D the macrophages are noted using MÏ• symbol, which is not consistent with the notation in the text where the same cell type appears as BMDM.
  • The last paragraph of the text of subchapter 2.3. is actually a conclusion, which isn't exactly well placed here.
  • For the detection of apoptosis in Jurkat cells (Figure 4B), STS-exposed Jurkat cells were compared with a blank DMSO sample. It would have been much more relevant to compare the apotosis levels of non-STS-exposed Jurkat cells with those exposed to STS.
  • The images in Figure 4D are not conclusive for the phenomenon of engulfing of apoptotic Jurkat cells by BMDM, it could actually be a simple overlap of these cells.

The text also shows some typos errors that obviously need to be corrected:

  • page 18, last line: "German" appears incorrectly instead of "Germany"
  • page 14, first paragraph of the discussion chapter, line 13: incorrectly appears “Annelise et al” instead of “Pezzi et al” (misuse of name instead of surname)
  • page 16, paragraph 4.4., 4th line: incorrectly appears “cutures”, instead of “cultures”

Reviewer 2 Report

Authors examined the effects of consecutive hypoxic culture on the proliferation and senescence of mesenchymal stem cells (MSCs), and the efficacy of MSCs secretome in the treatment of ALI using in vitro and in vivo experimental models. The contents of the present study are of great interests including detailed analysis of secretomes derived from normoxia-MSCs or hypoxia-MSCs with adequate experiments, which would provide a valuable information on the relevant research field, especially in development of therapeutic approach against acute lung injury. Minor comments in below may be considered.

In Introduction, why Authors focus on the senescence of MSCs should be logically described.

In clinic, any relevant studies to demonstrate that individual pulmonary circumstances like hypoxia or normoxia conditions affect the therapeutic efficacy of MSCs?

Although TGF-β is exactly anti-inflammatory cytokine, while it does strongly promote fibrotic foci mediating conversion of resident fibroblast into active myofibroblast. In the present study, authors focus on the macrophages, but not other important pulmonary compartments like epithelium and fibroblasts regarding the effect of secretomes derived from MSCs. Some comments about other compartments than macrophages should be needed considering the therapeutic effect of MSCs against acute lung injury.
